# Detection of Virulence-Associated Genes among *Brucella melitensis* and *Brucella abortus* Clinical Isolates in Greece, 2001–2022

**DOI:** 10.3390/pathogens12111274

**Published:** 2023-10-24

**Authors:** Joseph Papaparaskevas, Alexandra Procopiou, John Routsias, Georgia Vrioni, Athanasios Tsakris

**Affiliations:** Department of Microbiology, Medical School, National and Kapodistrian University of Athens, 75 M. Asias Str., 11527 Athens, Greece; ipapapar@med.uoa.gr (J.P.); alexandraproc17@hotmail.com (A.P.); jroutsias@med.uoa.gr (J.R.); atsakris@med.uoa.gr (A.T.)

**Keywords:** *Brucella melitensis*, *Brucella abortus*, virulence gene, pathogenicity

## Abstract

Brucellosis remains an important zoonotic disease in several parts of the world; in Greece, although it is declining, it is still endemic, affecting both the financial and public health sectors. The current study was undertaken to investigate the presence and distribution of virulence-associated genes among *Brucella* spp. clinical strains isolated during 2001–2022. Species identification was performed using conventional methodology and Bruce-ladder PCR. The presence of the virulence genes *mvi*N, *man*A, *wbk*A, *per*A, *omp*19, *ure*, *cbg* and *vir*B was investigated using PCR. During the study period, a total of 334 *Brucella* isolates were identified, of which 328 (98.2%) were detected from positive blood cultures; 315 (94.3%) of the isolates were identified as *B. melitensis*, whilst the remaining 16 (4.8%) and 3 (0.9%) were identified as *B. abortus* and *B. suis*, respectively. Notably, two of the *B. melitensis* were assigned to the REV-1 vaccine strain type. The presence of the *omp*19, *man*A, *mvi*N and *per*A genes was confirmed in all 315 *B. melitensis* isolates, while *ure*, *wbk*A, *cbg* and *vir*B genes were detected in all but 9, 2, 1 and 1 of the isolates, respectively. All eight virulence genes were amplified in all *B. abortus* and *B. suis* isolates. The detection rate of virulence genes did not differ significantly among species. In conclusion, brucellosis is still considered a prevailing zoonotic disease in Greece, with the majority of the isolates identified as *B. melitensis*. The eight pathogenicity-associated genes were present in almost all *Brucella* isolates, although the *ure* gene was absent from a limited number of *B. melitensis* isolates.

## 1. Introduction

Brucellosis is an endemic zoonotic disease in Greece [1,2]. The Hellenic National Public Health Organization (NPHO) indicated a mean annual notification rate of 1.3 cases per 100,000 per year during 2000–2022, with a decreasing trend (https://eody.gov.gr/wp-content/uploads/2019/05/brucellosis_epidemiologic_data_2010_2022.pdf, Accessed on 20 September 2023). Interestingly, an outbreak due to *Brucella melitensis* biovar 3 (from locally produced raw cheese) was documented during 2008 on the island of Thassos, Northern Greece [1]. Nevertheless, the limited available data for the last decade indicate a deteriorating situation (NPHO, personal communication), whilst comparison of locally obtained seroprevalence data indicates possible underreporting [2].

*Brucella* pathogenicity is mainly linked to its ability to survive and reproduce intracellularly in the host cells, thus evading the host immune reaction, rather than with the presence and/or production of exotoxins, fimbria, capsules, plasmids and endotoxic lipopolysaccharide. This is mainly facilitated by the expression of several molecules of the cell envelope, such as the integral membrane-bound protein (MviN), the mannose-6-phosphateisomerase (ManA), the mannosyl-transferase (WbkA), the perosamine synthetase (PerA) and the outer membrane protein 19 (Omp19), which contribute to the control of intracellular transport of the pathogen [3] and the initial survival of bacteria in macrophages and other cells of the reticuloendothelial system.

Furthermore, *Brucella* produces cyclic β(1–2) glucan (encoded by the *cbg* gene), a low-molecular-weight polysaccharide, also known as polysaccharide B, which interacts with lipids and contributes to pathogen survival by avoiding phagosome fusion with lysosomes, whilst the *vir*B gene regulates one of the most important virulence factors that secretes host cell effectors and regulators and contributes to the survival of bacterial cells inside host cells [4,5]. Finally, the *ure* gene is involved in the pathogenicity encoding the metalloenzyme urease production. The protease hydrolyzes urea to ammonia, raising the pH and thus enabling survival in acidic environments, such as the stomach [6].

All these virulence factors produce a micro-environment that facilitates survival of the bacterial cell and expression of its pathogenicity [3]. Thus, the presence of the respective genes in the *Brucella* genome confirms the pathogenicity of the microorganism.

Specific studies regarding investigation of the spread of the virulence factors among clinical *Brucella* isolates and potential differences between species are limited in the literature. In this respect, the present study was designed in order to investigate the presence of the *mvi*N, *man*A, *wbk*A, *per*A, *omp*19, *ure*, *cbg* and *vir*B virulence genes among *Brucella* spp. clinical isolates recovered from different geographic regions in Greece.

## 2. Materials and Methods

### 2.1. Bacterial Isolates

A total of 334 unique *Brucella* spp. clinical isolates (one per patient) were collected during the period 2001–2022 from patients admitted to regional hospitals in seven provinces of mainland Greece (Attica, Central Greece, Western Greece, Thessaly, Macedonia, Thrace and Epirus). The catchment population corresponded approximately to 40% of the total national population. Identification at the genus and species level was initially performed in clinical laboratories of the regional hospitals where the patients were admitted, following their own conventional methodologies and protocols. The isolates were then submitted to the reference laboratory, Department of Microbiology, Medical School, National and Kapodistrian University of Athens.

### 2.2. Identification 

In the reference laboratory, sub-culture of the isolates was performed in the Biosafety Level III containment unit on 5% sheep blood agar plates (Bioprepare, 19001, Keratea, Greece), followed by incubation at 36 °C in 5% CO_2_ conditions for 48 h. The genus was re-confirmed or assigned using conventional methodology and a previously described PCR protocol [7]. Species identification and differentiation of the wild-type and vaccine-type isolates (*B. melitensis*, *B. abortus*, *B. suis*, *B. canis*, vaccine *B. melitensis* REV-1 and vaccine *B. abortus* RB-51) was performed using the Bruce-ladder PCR protocol [8,9]; this multiplex protocol, using a set of eight pairs of primers, results in DNA band patterns that are unique for each species. 

### 2.3. Molecular Detection of Virulence Genes

Colonies from 48 h cultures were subjected to DNA extraction using the QIAamp DNA Mini Kit (Qiagen, Hilden, Germany). Detection of the presence and distribution of the pathogenicity genes *mvi*N, *man*A, *wbk*A, *per*A, *omp*19, *ure*, *cbg* and *vir*B was performed according to previously described protocols [3,10,11]. The target genes and the primer sequences are shown in Table 1. The PCRs were performed in two multiplex reactions (mixA, comprising the genes *omp*19, *mvi*N, *man*A, *per*A and *wbk*A, and *mix*B, comprising the genes *vir*B and *cbg*) and a single PCR reaction (for the *ure* gene). All PCR reactions were performed in 50 μL final volumes, using the GoTaq^®^ Green Master Mix (Promega Corp., Madison, WI 53711, USA) and a MyCycler thermal cycler (BioRad Laboratories MEPE, 11527, Athens, Greece). 

### 2.4. Reference Strains

Reference strains included the strain *B. melitensis* ATCC23456 (16M) and the vaccine strains *B. melitensis* REV-1 and *B. abortus* RB-51 (CZ Veterinaria SA, Porrino, 6400, Spain) obtained from the Animal Health Directorate of the Hellenic Ministry of Rural Development and Food. 

### 2.5. Statistical Evaluation

Statistical evaluation was performed using Pearson’s x^2^, and significant difference was set at *p* < 0.01 (indicating strong evidence).

## 3. Results and Discussion

The present study was conducted in order to investigate the presence and distribution of virulence factors among *Brucella* clinical isolates recovered in Greece.

The majority of the isolates that were studied (328/334; 98.2%) were derived from positive blood cultures, whilst three (0.8%), one (0.3%), one (0.3%) and one (0.3%) of the isolates were obtained from bone marrow, vertebral biopsy, synovial fluid and cerebrospinal fluid cultures, respectively. The Bruce-ladder protocol identified 315 (94.3%) of the isolates as *B. melitensis*, whilst 16 (4.8%) isolates were identified as *B. abortus* and 3 (0.9%) as *B. suis*. In addition, the Bruce-ladder PCR protocol differentiated two of the *B. melitensis* isolates from the wild-type strains and assigned them to the REV-1 vaccine type.

In this respect, the study confirmed existing data from previous national reports regarding the dominance of *B. melitensis*, as compared to the scarce isolation of *B. abortus* [1,2]. It is noteworthy to mention that this is the first report of *B. suis* isolation from clinical specimens in our country. *B. suis* is very infrequent and, despite a few recorded outbreaks, the European Union countries are considered officially free of porcine brucellosis [12], although older reports from the Balkan region indicated its sporadic isolation [13]. However, *B. suis* (especially biovar 2) is widespread in wildlife, its reservoir being wild boars and brown hares [12]. In this respect, as these wild animals are frequent in Northern Greece (where these two clinical cases were detected), such a transmission cannot be excluded.

The Bruce-ladder PCR identification protocol, although technically demanding, is currently considered the standard molecular method of identifying *Brucella* spp. to the species level [8,9] and in addition can discriminate the main vaccine types (REV-1, RB-51 and S19). This was the case with two of our *B. melitensis* isolates, indicating possible exposure during the animal vaccination process.

It should also be noted that both molecular protocols used in this study are able to differentiate between the clinically important *Brucella* spp. (*melitensis*, *abortus*, *suis*, *canis*) and the *Ochrobactrum* spp. that have been recently reclassified together with the *Brucella* spp. (https://www.cdc.gov/locs/2022/12-19-2022-Lab-Update-Reclassification_Ochrobactrum_species_Brucella_genus.html and https://asm.org/getmedia/ae7fc7c3-0281-43aa-9143-3c9dae22a85e/Brucella-and-Ochrobactrum-Taxonomic-Updates-for-Laboratories.pdf?ext=.pdf, both accessed on 20 September 2023), indicating differences between the two group of species, although much scientific dispute has arisen regarding this classification [14].

Most of the isolates in our survey harbored the genes expressing the cell envelope factors, which constitute the main virulence elements of the *Brucella* genus. More specifically, the virulence genes *omp*19, *man*A, *mvi*N and *per*A were detected in all 315 *B. melitensis* isolates, whilst *ure*, *wbk*A, *cbg* and *vir*B genes were detected in all but 9, 2, 1 and 1 of the isolates, respectively. The various detection patterns among the *B. melitensis* isolates are depicted in Table 2. All eight virulence genes were amplified in the 16 *B. abortus* and 3 *B. suis* isolates in the study. The detection rate of virulence genes did not differ significantly among *Brucella* species.

Similar results have been obtained from recent studies from Iran, Egypt and Greece using whole-genome sequencing [15,16,17]. These studies showed that the majority of isolates do possess the virulence genes under discussion and there are limited (if any at all) differences between species. It should be noted that although our study did not use WGS but specific PCRs, the results were comparable. Nevertheless, the value of WGS should not be underestimated, as it may provide a significant amount of data in a short period of time, data that may not be visible from the beginning, thus potentially outperforming specific PCR protocols.

These findings are in line with the fact that all the isolates were derived either from positive blood cultures (most of them) or from other invasive infections (bone marrow, vertebral biopsy, synovial fluid, and cerebrospinal fluid), thus confirming their pathogenic status. The only noteworthy difference (although not statistically significant) between the two species was regarding the *ure* gene, which was not detected in 9 (2.9%) of the 315 *B. melitensis* isolates, all of which, however, were also isolated from positive blood cultures. It should be noted that the number of *B. abortus* strains in our collection was limited, due to the prevalence of the *B. melitensis* species in Greece. But even in these few strains, all eight pathogenicity genes were detected, in contrast to *B. melitensis* strains.

Urease is a nickel-containing enzyme that catalyzes the hydrolysis of urea to ammonia, which in turn is being used as a nitrogen source by bacteria. In addition, ammonia release increases the acidic pH of the stomach, which is one of the most important barriers of the gastrointestinal tract, thus enabling pathogens to survive and move to the intestine. Furthermore, urease has been identified as an immunogenic regulator in various pathogen-induced inflammatory responses, suggesting that it has a role in pathogenesis independent of its endogenous enzymatic activity [6].

The study by Sangari et al. [6] indicated that most of the *Brucella* species express strong urease activity, and this activity is responsible for their ability to survive through transmission of infection through the gastrointestinal tract. This finding is also supported by the fact that the *B. ovis* species, which lacks urease activity, is unable to cause human infection through the gastrointestinal tract.

In the present study, a total of 2.9% of the *B. melitensis* isolates were found not to harbor the *ure* gene, although these isolates had proven pathogenic capability since they were derived from positive blood culture from hospitalized patients with clinical manifestations. Although the point of entry is not known for every single patient, it is safe to assume that it most probably was through the gastrointestinal tract, thus suggesting that the isolates possessed the required ability to survive the acidic stomach environment. These results agree with a previous study [18] indicating that not all of the *Brucella* clinical isolates express urease activity. In addition, it should be noted that certain laboratory-processed strains, such as the *B. abortus* 544, are urease-negative, although they are both pathogenic and capable of infection [6].

Regarding the geographical distribution, there were no significant differences, and the *ure*-negative isolates were detected in various regions of the country.

Regarding the single isolate that was found to be negative for the *cbg* and *vir*B genes, as well as the two isolates that were found to be negative for the *wbk*A gene, all these isolates were also derived from positive blood cultures. In general, these genes are detected in the majority of strains of human and/or animal origin. Nevertheless, there are also studies indicating that as many as 8% of the isolates may actually be lacking some or all of these genes [11], although their pathogenicity does not seem to be questionable.

Based on these data, we could assume that the properties of the urease enzyme, as well as its importance in the metabolism or pathogenicity of *Brucella*, have not been fully elucidated. Further studies are needed to define differences in pathogenicity between *ure*-positive and *ure*-negative *Brucella* strains and whether urease’s activity is actually complementary to the pathogenic activity of the cell envelope virulence factors. In addition, further studies are needed in order to elucidate the actual effects of the absence of other genes on the pathogenic capability of the strains.

The main limitations of the present study are as follows: (1) the nature of the reference of the isolates was voluntary and ad hoc, thus preventing the extrapolation of the results for the whole country, (2) an appropriate comparison between *Brucella* species was not possible due to the limited number of *B. abortus* isolates detected, and (3) our methodology only detected the presence and not the functionality of the genes under investigation.

## 4. Conclusions

In conclusion, the current study confirmed previous reports that *B. melitensis* is the dominant *Brucella* species in Greece. It has also shown that all eight investigated virulence genes contributing to *Brucella* pathogenicity were present in the vast majority of the tested isolates. Exceptionally, the *ure* gene was not amplified from a limited number of *B. melitenis* isolates. Three other virulence genes (*wbk*A, *cbg* and *vir*B) were also not detected in very few (two or one) of the *B. melitensis* isolates.

## Figures and Tables

**Table 1 pathogens-12-01274-t001:** Sequences, amplicon size and references of the primers used in the present study.

Gene	Sequence	Amplicon (bp)	Reference
*omp*19	F: 5′-TGATGGGAATTTCAAAAGCA-3 R: 5′-GTTTCCGGGTCAGATCAGC-3′	550	[3]
*wbk*A	F: 5′-AATGACTTCCGCTGCCATAG-3′ R: 5′-ATGAGCGAGGACATGAGCTT-3′	931	[3]
*man*A	F: 5′-TCGATCCAGAAACCCAGTTC-3′ R: 5′-CATACACCACGATCCACTGC-3′	271	[10]
*mvi*N	F: 5′-GCAGATCAACCTGCTCATCA-3′ R: 5′-GGCCATAGATCGCCAGAATA-3′	344	[3]
*ure*	F: 5′-GCTTGCCCTTGAATTCCTTTGTGG-3′ R: 5′-ATCTGCGAATTTGCCGGACTCTAT-3′	2100	[11]
*per*A	F: 5′-GGAACGGTGGCACTACATCT-3′ R: 5′-GGCTCTCTGTGTTCCGAGTT-3′	716	[3]
*cbg*	F: 5′- GAATTCGCCAATGAGGAAAA-3′ R: 5′- ACGATATCGGATGCGAAAAG-3′	575	[10]
*vir*B	F: 5′- CGCTGATCTATAATTAAGGCTA -3′ R: 5′- TGCGACTGCCTCCTATCGTC-3′	881	[10]

**Table 2 pathogens-12-01274-t002:** Detection patterns of the eight virulence genes among the 315 *B. melitensis* isolates.

*B. melitensis* (No)	*omp*19	*wbk*A	*man*A	*mvi*N	*ure*	*per*A	*cbg*	*vir*B
305	+	+	+	+	+	+	+	+
7	+	+	+	+	Negative	+	+	+
2	+	Negative	+	+	Negative	+	+	+
1	+	+	+	+	+	+	Negative	Negative

## Data Availability

All PCR photos are available on request.

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
