# Peer review of "Detection of Virulence-Associated Genes among Brucella melitensis and Brucella abortus Clinical Isolates in Greece, 2001–2022"

_pathogens, 2023, doi:10.3390/pathogens12111274_

Round 1

Reviewer 1 Report (New Reviewer)

The manuscript ” Detection of virulence-associated genes among Brucella melitensis and Brucella abortus clinical isolates in Greece, 2001-2022 " submitted for publication in PATHOGENS is of great interest to me and Brucellosis researcher. The following are points that should be addressed for acceptance

1.     What was the possible reason for the isolation of  REV-1 vaccine strain type isolates in this study, line 28

2.     Provide PCR thermal conditions used for each virulent gene amplification

3.     Have you sequenced amplified product from each virulent gene for verification using BLAST analysis

4. 

Author Response

The manuscript ”Detection of virulence-associated genes among Brucella melitensis and Brucella abortus clinical isolates in Greece, 2001-2022 " submitted for publication in PATHOGENS is of great interest to me and Brucellosis researcher. The following are points that should be addressed for acceptance.

  1. What was the possible reason for the isolation of  REV-1 vaccine strain type isolates in this study, line 28.

As already mentioned in the discussion section (second paragraph, lines 167-168 of the original submission) we believe that it was due to “…possible exposure during the animal vaccination process”.

  1. Provide PCR thermal conditions used for each virulent gene amplification.

The conditions were exactly the same as in the reference provided, as mentioned in lines 97-98 of the original submission. In order to save space (and since the manuscript is in the short communication format), we decided not to replicate this information.

  1. Have you sequenced amplified product from each virulent gene for verification using BLAST analysis.

No, unfortunately we have not sequenced the amplicons.

Reviewer 2 Report (New Reviewer)

Abstract:

·        L19: important zoonosis…………..  ………..important zoonotic disease…..

·        L 20 : Although declining………..  Although declining status

Introduction:

·        L66: please add reference.

Materials and methods:

·        Where is the ethical commitment for this study?? Although this study was done using isolates from over 20 years. But ethical approval for collection of samples or even conducting the current study should be obtained.

·        L73: please clarify the types of patient samples.

·        L84: What about the preservation condition of these isolates for more than 20 years? I am afraid that is a very long period.

·        Did these isolates were identified previously during 2001?

·        Unfortunately, no link between the obtained results and the risk factors during the time of collection which could make the study more worthy.

·        My suggestion for the authors is to obtain new isolates from currently new patients and compare between the old and the new isolates.

Discussion:

L178-188: is an obtained result should be at the result section? Kindly, where is the discussion of these results?

Author Response

Abstract:

  • L19: important zoonosis…………..  ………..important zoonotic disease…..

It has been corrected.

  • L 20 : Although declining………..  Although declining status.

It has been corrected.

Introduction:

  • L66: please add reference.

It has been added (ref. #3)

Materials and methods:

  • Where is the ethical commitment for this study?? Although this study was done using isolates from over 20 years. But ethical approval for collection of samples or even conducting the current study should be obtained.

In Greece there is no need for specific ethical approval for studying archived microbiological material (strains) of one or two decades old. In addition, the strains were isolated during the everyday clinical work of the respective labs and part of this work (like the conventional and the molecular species identification, etc.) was performed in the line of this clinical work.

  • L73: please clarify the types of patient samples.

The samples have been clarified in results section (lines 137-140 of the original submission). More specifically: “The majority (328/334; 98.2%) were derived from blood cultures, whilst three (0.8 %), one (0.3 %), one (0.3 %) and one (0.3%) isolates were obtained from bone marrow, vertebral biopsy, synovial fluid and cerebrospinal fluid cultures, respectively.”

  • L84: What about the preservation condition of these isolates for more than 20 years? I am afraid that is a very long period.

Preservation of the isolates was performed in deep refrigeration (-80o C) with periodical subculture every few years in order to check viability. Preservation of the DNA was also done in deep refrigeration.

  • Did these isolates were identified previously during 2001?

As mentioned in materials and methods (lines 77-79 of the original submission): “Identification at the genus and species level was initially performed in clinical laboratories of the regional hospitals where the patients were admitted, following their own conventional methodologies and protocols.” That means the initial identification was performed during isolation.

  • Unfortunately, no link between the obtained results and the risk factors during the time of collection which could make the study more worthy.

This is a very good point. Unfortunately, we are in possession of only basic information regarding the isolates (name, gender, specimen type, area of the country, species identification). In that respect we cannot get a definite conclusion or even speculate regarding possible risk factors.

  • My suggestion for the authors is to obtain new isolates from currently new patients and compare between the old and the new isolates.

We would like to clarify that the isolates presented here were not just from 2001. They were obtained during the whole of the period 2001-2022. A significant number of these isolates were in fact recent (from years 2022, 2021, 2020, 2019, etc.).

Discussion:

L178-188: is an obtained result should be at the result section? Kindly, where is the discussion of these results?

The results and discussion sections have been combined in the revised version (as per suggestions from other reviewer), in that respect this issue has been sorted out.

Reviewer 3 Report (New Reviewer)

The article by Papaparaskevas et al., which discusses the detection of virulence-associated genes among Brucella isolates in Greece, is interesting, original, and suitable for publication in this journal. The article is well-written and covers an important aspect of bacterial zoonotic brucellosis. The authors analyzed clinical strains isolated over 20 years from clinical cases. They used PCR to confirm the presence of the virulence genes mviN, manA, wbkA, perA, omp19, ure, cbg and virB in 334 isolates. Despite the final conclusion presented here was provided before in several studies in other countries, However, using a different set of isolates from humans over a long time period is a great job. The difference in the distribution of virulence-associated genes in all tested brucella species is insignificant. This has been provided in several studies before using WGS technology. Please refer to some studies in the discussion. Some comments need to be addressed before publication as the following:

·        The introduction is weak and can be improved by adding more information about the current situation of brucellosis in Greece and neighboring countries in the Mediterranean basin; you may check and cite the following studies: DOI: 10.1186/s12917-022-03295-4 Doi: https://doi.org/10.51585/gjvr.2022.1.0037 , https://doi.org/10.3390/microorganisms9091942.

·        Few studies have been carried out on determining Virulence-Associated Genes in Brucella spp.; please highlight some of them in the introduction, such as a study from Iran doi: 10.3390/pathogens12010082 and Egypt doi: 10.1016/j.onehlt.2021.100255 . The difference in the distribution of 43 known virulence-associated genes in all tested brucella species is insignificant. This has been provided in both studies before using WGS technology. Please refer to both studies in the discussion.

·         Recently, a novel study discussed the genotype diversity of brucellosis agents isolated from humans and animals in Greece based on whole-genome sequencing. The study also highlighted the possible differences in virulence genes. Please discuss your results compared to this study DOI: 10.1186/s12879-023-08518-z.

·        Please mention the total number of isolates in the material and method section.

·        Authors mentioned that “15 Greek hospitals located in 8 seven different geographic regions (Attica, Central Greece, Western Greece, Thessaly, Macedonia, Thrace, and Epirus), it is 8 or 7, please correct and include all information in materials.

·        I suggest making a table in the results including location, Brucella species, and source of samples

·        I recommend combining of results and discussion as one section.

·        Please discuss the value of using WGS in the detection of all virulence genes in comparison to PCR which is detect only target genes with known primers

·        Classification of Brucella together with Ochrobactrum was a big mistake, and the pioneers of Brucella published already articles to prove that both bacteria must be separated. Please remove this part as IT IS A MISTAKE from the discussion and include those two articles

1.      J Clin Microbiol . 2023 Aug 23;61(8):e0043823. doi: 10.1128/jcm.00438-23. Epub 2023 Jul 3. If You're Not Confused, You're Not Paying Attention: Ochrobactrum Is Not Brucella

2.      Genetic comparison of Brucella spp. and Ochrobactrum spp. erroneously included into the genus Brucella confirms separate genera. Ger. J. Vet. Res 2023. vol. 3, Iss. 1 pp:31-37. Doi: https://doi.org/10.51585/gjvr.2023.1.0050

Little English editing is required to improve the text

Author Response

The article by Papaparaskevas et al., which discusses the detection of virulence-associated genes among Brucella isolates in Greece, is interesting, original, and suitable for publication in this journal. The article is well-written and covers an important aspect of bacterial zoonotic brucellosis. The authors analyzed clinical strains isolated over 20 years from clinical cases. They used PCR to confirm the presence of the virulence genes mviN, manA, wbkA, perA, omp19, ure, cbg and virB in 334 isolates. Despite the final conclusion presented here was provided before in several studies in other countries, However, using a different set of isolates from humans over a long time period is a great job. The difference in the distribution of virulence-associated genes in all tested brucella species is insignificant. This has been provided in several studies before using WGS technology. Please refer to some studies in the discussion. Some comments need to be addressed before publication as the following:

  • The introduction is weak and can be improved by adding more information about the current situation of brucellosis in Greece and neighboring countries in the Mediterranean basin; you may check and cite the following studies: DOI: 10.1186/s12917-022-03295-4 (only from clinical specimens from animals, no isolates), Doi: https://doi.org/10.51585/gjvr.2022.1.0037 (all isolates from animals, https://doi.org/10.3390/microorganisms9091942 (majority of isolates from animals, only two from human).

We agree with the reviewer that these (among others) are additional references that can be included in the introduction. Nevertheless, the article is in the short communication format, the number of references that can be included is not high and has already been surpassed (taking also into account the references to be added according to the following comments). In that respect, we decided to include only references from studies comprising clinical strains and not strains from animal origin (as this is more relevant to the study).

  • Few studies have been carried out on determining Virulence-Associated Genes in Brucella spp.; please highlight some of them in the introduction, such as a study from Iran doi: 10.3390/pathogens12010082and Egypt doi: 10.1016/j.onehlt.2021.100255 . The difference in the distribution of 43 known virulence-associated genes in all tested brucella species is insignificant. This has been provided in both studies before using WGS technology. Please refer to both studies in the discussion.

We agree with the reviewer. These two studies have been included in the reference section and a short paragraph has been included in the combined results-discussion section.

  • Recently, a novel study discussed the genotype diversity of brucellosis agents isolated from humans and animals in Greece based on whole-genome sequencing. The study also highlighted the possible differences in virulence genes. Please discuss your results compared to this study DOI: 10.1186/s12879-023-08518-z.

We agree with the reviewer. This study has been included in the references and a short paragraph has been included in the combined results-discussion section.

  • Please mention the total number of isolates in the material and method section.

The total number (n=334) has been included in the materials and method section.

  • Authors mentioned that “15 Greek hospitals located in 8 seven different geographic regions (Attica, Central Greece, Western Greece, Thessaly, Macedonia, Thrace, and Epirus), it is 8 or 7, please correct and include all information in materials.

Thank you for pointing out this typo error. It has been corrected and included in the materials section.

  • I suggest making a table in the results including location, Brucella species, and source of samples

We agree that a table might help in better presenting these results, but the article is in the short communication format, and it is not permitted to include any more tables, apart from the ones already submitted.

  • I recommend combining of results and discussion as one section.

Thank you for this suggestion, the two sections have been combined.

  • Please discuss the value of using WGS in the detection of all virulence genes in comparison to PCR which is detect only target genes with known primers

A short paragraph has been added in the discussion section, together with the WGS studies mentioned in the previous comments.

  • Classification of Brucella together with Ochrobactrum was a big mistake, and the pioneers of Brucella published already articles to prove that both bacteria must be separated. Please remove this part as IT IS A MISTAKE from the discussion and include those two articles
  1. J Clin Microbiol . 2023 Aug 23;61(8):e0043823. doi: 10.1128/jcm.00438-23. Epub 2023 Jul 3. If You're Not Confused, You're Not Paying Attention: Ochrobactrum Is Not Brucella
  2. Genetic comparison of Brucella spp. and Ochrobactrum spp. erroneously included into the genus Brucella confirms separate genera. Ger. J. Vet. Res 2023. vol. 3, Iss. 1 pp:31-37. Doi: https://doi.org/10.51585/gjvr.2023.1.0050

The reviewer is correct about the scientific dispute that has been initiated since last year’s reclassification of Ochrobactrum and Brucella species.  It should be noted however that in our study we did not try to address this particular issue, we just pointed out that the PCR-based identification methodology used was valid for differentiating the clinically relevant brucella species from the Ochrobactrum species, and that in our collection no Ochrobactrum spp. isolate was included. In that respect, this in fact is in favor of the reviewer’s point of view (we have rephrased the sentence accordingly). Given the fact that the situation is still under discussion and no backwards reclassification has occurred (splitting of the two groups that have been merged), our opinion is that this phrase should remain, indicating differences between the two group of species. The JCM reference has been included, as suggested.

 Comments on the Quality of English Language Little English editing is required to improve the text.

We have checked throughout the manuscript for proper English.

Reviewer 4 Report (New Reviewer)

The manuscript by Joseph Papaparaskevas and colleagues is a report on the occurrence of virulence-associated genes in Brucella melitensis, B. abortus and B. suis clinical isolates. Overall, is a clear, concise, and well-written manuscript. The introduction is relevant and sufficient information about the previous study findings is presented for readers to follow the present study. In general, the research is well described, and the conclusions are supported by the analysis of the data presented. However, some points should be improved for clarity. Comments and suggestions are given below:

Title:

I understand that only one B. suis isolate was included in this study. However, this is also the first time that B. suis has been isolated from human isolates in Greece. My recommendation is to include B. suis also in the title.

Materials and Methods:

- “Bacterial isolates” section: from which samples isolates were obtained? it should be briefly described the culture conditions for the primary isolation.

- “Statistical evaluation” section: A reference for the statistical method applied should be given. For clarification, indicate that significant difference at p<0.01 means a “strong evidence”.

- Lines 66 and 68: “Brucella” should be in italic.

- Line 101: “ure” should be in italic.

Discussion section:

-  Lines 154-156: Please revise the sentence “It should be noted however that this is the first report of B. suis isolation from clinical specimens in our country.”  One possibility: “It is noteworthy to mention that this is the first report of B. suis isolation from clinical specimens in our country”.

The manuscript is written in an efficient manner and without major errors. Minor editing of English language is required.

Author Response

Comments and Suggestions for Authors

The manuscript by Joseph Papaparaskevas and colleagues is a report on the occurrence of virulence-associated genes in Brucella melitensisB. abortus and B. suis clinical isolates. Overall, is a clear, concise, and well-written manuscript. The introduction is relevant and sufficient information about the previous study findings is presented for readers to follow the present study. In general, the research is well described, and the conclusions are supported by the analysis of the data presented. However, some points should be improved for clarity. Comments and suggestions are given below:

Title:

I understand that only one B. suis isolate was included in this study. However, this is also the first time that B. suis has been isolated from human isolates in Greece. My recommendation is to include B. suis also in the title.

Thank you for this comment. The aim of this study is to compare potential differences between B. melitensis and B. abortus isolates regarding the virulence factors. The three B. suis strains that have been isolated, although a notable discovery, is only a side result of the main study, that should be mentioned, but (in our opinion) does not merit inclusion in the title, which is already long enough. 

Materials and Methods:

- “Bacterial isolates” section: from which samples isolates were obtained? it should be briefly described the culture conditions for the primary isolation.

The clinical specimens are depicted in results section (lines 137-140 of the original submission). More specifically: “The majority (328/334; 98.2%) were derived from blood cultures, whilst three (0.8 %), one (0.3 %), one (0.3 %) and one (0.3%) isolates were obtained from bone marrow, vertebral biopsy, synovial fluid and cerebrospinal fluid cultures, respectively.”

In addition, the culture conditions of the primary isolation were in accordance with the different laboratory protocols of the hospitals that referred the isolates (it is mentioned in lines 78-79 of the original submission). In that respect (and since the hospitals were numerous) we do not have all the information and in addition we do not have the necessary space to fully present the available details.

- “Statistical evaluation” section: A reference for the statistical method applied should be given. For clarification, indicate that significant difference at p<0.01 means a “strong evidence”.

The words “strong evidence” has been included in the sentence, as suggested. In addition, it has been clarified that the methodology was Pearson’s Chi-square two-by-two table.

- Lines 66 and 68: “Brucella” should be in italic.

It has been corrected.

- Line 101: “ure” should be in italic.

It has been corrected.

Discussion section:

-  Lines 154-156: Please revise the sentence “It should be noted however that this is the first report of B. suis isolation from clinical specimens in our country.”  One possibility: “It is noteworthy to mention that this is the first report of B. suis isolation from clinical specimens in our country”.

It has been rephrased as suggested.

Comments on the Quality of English Language

The manuscript is written in an efficient manner and without major errors. Minor editing of English language is required.

We have checked throughout the manuscript for proper English.

Round 2

Reviewer 3 Report (New Reviewer)

The authors addressed all required comments and the manuscript has been improved

This manuscript is a resubmission of an earlier submission. The following is a list of the peer review reports and author responses from that submission.

Round 1

Reviewer 1 Report

The authors investigated the abundance of virulence genes in Brucella spp. isolated from some clinics in Greece. As expected, the probed virulence genes were detected in the majority of isolates. While this may be out of scope of the study, it would be interesting to assess the absence of ure in some of the pathogenic isolates. 

Minor comment:

A brief description of the Bruce-Ladder PCR protocol in the materials and methods would provide more relevance to how this technique can distinguish between Brucella spp. 

Reviewer 2 Report

The manuscript is descriptive in nature and adds useful regional data.  The entire manuscript needs to be critically updated to review, account for and discuss the impact of reclassification Ochrobactrum species into the Brucella genus and to document any efforts by their laboratories to distinguish Brucella (Ochrobacterum) from select agent Brucella species to eliminate possible misidentifications.  See Centers for Disease Control and Prevention. 2022 .12/19/2022. Lab update: Reclassification of Ochrobactrum  species into the Brucella genus (cdc.gov). Accessed 14 February 2023. Also see https://asm.org/getmedia/ae7fc7c3-0281-43aa-9143-3c9dae22a85e/Brucella-and-Ochrobactrum-Taxonomic-Updates-for-Laboratories.pdf?ext=.pdf for more information and applicable references that will be needed to update the manuscript.

Reviewer 3 Report

The research is of interest and the manuscript is well prepared. I recommend including the wbkA, cbg and virB genes in the discussion. Is there any more information about patient’s anamnesis of the corresponding bacteria which were negatively tested for this virulence genes – was any effect observed?

Furthermore, I recommend reporting the limitations of the study as for instance that the method cannot prove for functional genes (genes could be not complete) or the other way around that the negative tested genes might be caused by base substitutions in the primer binding sites what not necessarily mean that the gene is absent or ineffective.

Line 44: “…caused by locally produced raw cheese.” I recommend rewriting as the causative agent was Brucella.

Round 2

Reviewer 2 Report

The authors describe isolation and identification of Brucella from patients and characterize virulence factors, but fail to discuss the significance and any application of the findings. How do the data significantly add to the body of knowledge of disease mechanisms, prevention, diagnostics or therapeutics of brucellosis?